# Resolving Ambiguities in SHARAD Data Analysis Using High-Resolution Digital Terrain Models

**Léopold Desage [1,\*], Alain Herique [1], Sylvain Douté [1], Sonia Zine [1] and Wlodek Kofman [1,2]**

[1] Université Grenoble Alpes, CNRS, CNES, IPAG, 38000 Grenoble, France
[2] Centrum Badan Kosmicznych Polskiej Akademii Nauk (CBK PAN), Bartycka 18A, PL-00–716 Warsaw, Poland
[\*] Correspondence: leopold.desage@univ-grenoble-alpes.fr

**Abstract:** The SHAllow RADar (SHARAD) onboard Mars Reconnaissance Orbiter (MRO) is a 20MHz Synthetic Aperture Radar (SAR) that probes the first hundreds of meters of the Martian subsurface. In order to interpret the detection of subsurface interfaces with ground penetrating radars, simulations using Digital Terrain Models (DTM) are necessary. This methodology paper focuses on the analysis of the first tens of meters of the Martian subsurface with SHARAD, comparing the use of different high-resolution DTMs for radar simulation, namely, from the High-Resolution Stereo Camera (HRSC) onboard the Mars Express and from the Context Camera (CTX) onboard MRO. The region of Terra Cimmeria was chosen as a demonstration area. It is a highly cratered southern midlatitude region, where, as will be discussed, the higher resolution of the aforementioned terrain models is mandatory to describe the surface at an acceptable level of detail for shallow subsurface radar interpretation. With a DTM corrected by photoclinometry using CTX imagery, we show that a reflector that was visible on SHARAD data but not on the simulation made with an HRSC DTM is, in fact, a surface echo that was not reproduced by the HRSC surface model. We also show that, unlike laser altimetry DTMs, optical DTMs are prone to artifacts that can make radar analysis more complicated for some scenarios. Reciprocally, we show that the comparison between radar and its corresponding simulated data is a way of assessing a DTM's quality, which is especially useful in missions where ground control points are lacking, unlike Martian observations.

**Keywords:** radar simulation; coherent simulation; high-resolution DTM; optical DTM; DTM quality estimation; SHARAD

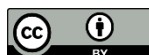

## 1. Introduction

The Martian subsurface has been studied with radars in orbit since the arrival and later activation of MARSIS in 2005. With its help and, later, with the use of the SHAllow RADar (SHARAD), water ice has been identified in the Martian subsurface, mainly at the poles [1,2] but also in the midlatitudes [3,4]. Both the ice characterization and an estimation of its quantity can be performed using radar data coupled with subsurface composition models. They are used to understand Mars's climatic past and to estimate the ice quantity for future in situ usage. The current climatic conditions at Mars's midlatitudes do not allow water ice to be stable at the surface. The detected ice in the midlatitudes is thought to have been deposited during past high-obliquity periods [5] and to have been covered by dust and sediments, which conserved it until today. The characterization and quantification of midlatitude subsurface ice allows us to understand the conditions that lead to its formation, and radars are a powerful tool to study those subsurface echoes. In order to detect subsurface interfaces with radars in obit, we need to eliminate the echoes coming from the surface. In fact, as the design of most radars in orbit provides them with a large antenna lobe, off-nadir echoes that could arrive with the same delay as a subsurface nadir echo are sensed by the radar. The classical method to remove this so-called

"clutter" consists of comparing the radar signal to simulations of the surface echoes using Digital Terrain Models (DTMs) [6].

This study focuses on simulations for the interpretation of data acquired by the SHAllow RADar onboard MRO. SHARAD is a synthetic aperture radar (SAR) with a wavelength of 15 m in free space and a Fresnel zone after Doppler compression of 300 m by 3 km horizontally [7]. The bulk of the signal sensed by the instrument comes from structures that are a non-negligible fraction of the Fresnel zone, but structures down to a fraction of the radar wavelength contribute to the signal. In order to properly simulate the signal perceived by SHARAD, we need, in theory, surface models that are at a resolution of around a dozen meters or better. In practice, most of the DTMs used for SHARAD simulations are generated from the MOLA instrument and are around 400 m in horizontal resolution. To discriminate deep (in the order of 150 m or deeper) and bright reflectors from the surface, MOLA products are sufficient to obtain satisfying results. When studying the first tens of meters of the subsurface in the Martian southern midlatitudes where the roughness is often high, MOLA products no longer meet the criteria, and we need higher resolution DTMs. This is mainly due to the high sensitivity to slope errors near Nadir, where the shallow echoes are detected.

The study aims at improving the detection of shallow subsurface near-nadir reflectors in challenging and rough terrain, with simulations of the SHARAD signal. In other words, the goal is to reproduce the original data as faithfully as possible. To achieve that, we need high-resolution surface models that reproduce the Martian surface at a sufficient resolution, with the lowest possible level of artifacts. In this paper, we present the simulations on different high-resolution DTMs, comparing the optical and laser altimetry terrain models and their impact on the radar signal reconstruction. This study also shows that the comparison between simulated and real radar data can be a way to assess DTM quality, which will be useful in missions where DTM sources are limited.

## 2. SHARAD Data Analysis with MOLA

This section will present the tools used for the simulations of SHARAD data as well as the area of interest chosen for this study. We also detail the first result on this area with a simulation using a MOLA DTM and discuss the limitations of those models.

### 2.1. SPRATS: A Toolset for Radar Simulation and Processing

SHARAD provides us with near-global coverage of the Martian surface and the first hundreds of meters of its subsurface. To perform the radar simulations needed for SHARAD data interpretation, we used SPRATS [8], which is a toolset developed at IPAG/Université Grenoble Alpes that allows for physical optics radar simulations [6,9]. This tool takes the trajectory of a given radar along with a model of the surface and simulates the echoes that are sensed by the radar pulse by pulse (Figures 1 and 2a). This coherent simulator uses the Huygens–Fresnel principle to compute the total signal received by the radar at every pulse, therefore taking into account the diffraction effects and coherently summing the contribution of every facet. This is the main difference with the more widely used incoherent radar simulators, where an incoherent summing of the echoes is achieved. Our method results in a less continuous and more "grainy looking" output result (Figure 2b). The other difference with incoherent simulators is that the echo power is summed on every facet of the DTM for every pulse (with a distance threshold), whereas for incoherent simulators, it is performed only across-track. The incoherent summing has the advantage of being lighter in terms of computing load, but a coherent summing method allows for an output result that is closer to the raw signal effectively measured by the radar. In fact, the simulated signal is then fed into an integrated SAR processor with the capacity of achieving synthesis on any 3-dimensionnal surface. The SAR synthesis of a SHARAD simulation using a MOLA DTM at a 463m horizontal resolution is shown Figure 2b. The main advantage of this simulation method is that the raw signal is simulated, and a SAR processing is applied with the same parameters as the original radargrams

processed by the US team (products referred as USRDR in the PDS), namely, the integration time and the multilook setting. This ability for detailed processing allows for a better fine details analysis, as well as true power comparison capability. Our aim with this clutter simulator is being as close as possible to the original SHARAD data.

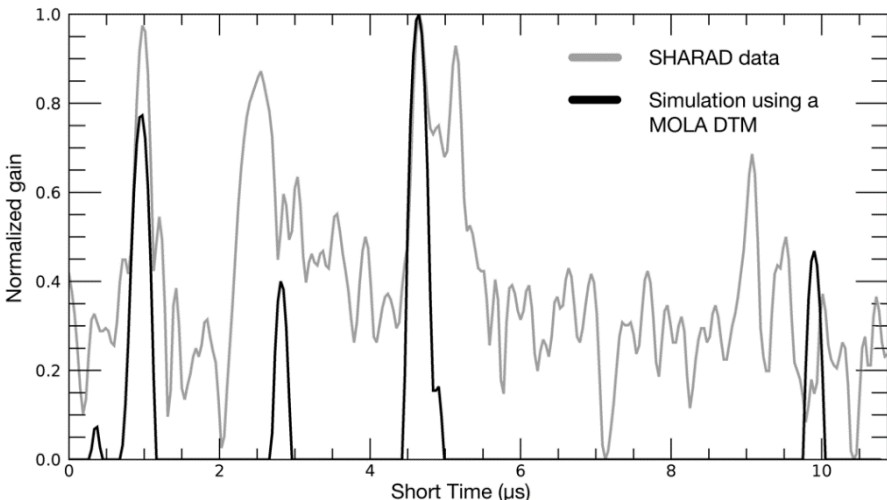

**Figure 1.** Single echo of SHARAD data compared to an echo of the simulation. The largest peaks of SHARAD data are well matched to the simulation.

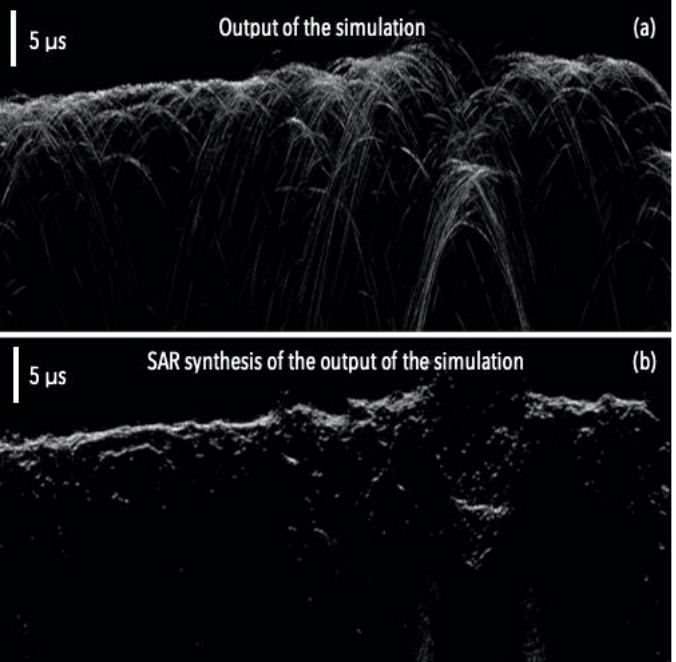

**Figure 2.** (**a**) Output of the simulation of a 150 km portion of the SHARAD dataset n°5128501 in Terra Cimmeria using a MOLA DTM. (**b**) Same profile after SAR synthesis. Parabolas in the simulation are the result of the trajectory of the spacecraft relative to each reflector.

### 2.2. Region of Interest

Terra Cimmeria is a southern midlatitude region of high rugosity. Composed of rugged terrain and a highly cratered surface, this region contains shallow subsurface reflectors visible on SHARAD data and not in MOLA simulations. Adeli et al. [10] analyzed Terra Cimmeria, more specifically, the ice deposits called valley fill deposits (VFD) around Tarq crater (−38.1°; 171.2°), to estimate whether they are ejecta deposits or formed by precipitations. During their study, they performed a simulation of SHARAD dataset

n°5128501 using a HRSC DTM and identified a reflector that is present on the SHARAD data, but not on their simulation. Their geomorphological study and the size of the reflector did not allow them to distinguish between the hypothesis of it being a surface or a subsurface echo. With a preliminary look at high-resolution images and by estimating the delay of the identified echo (as will be discussed later in Section 4.2), we have good reasons to think that the echo indeed comes from the surface and that a higher resolution DTM might be able to resolve the ambiguity.

### 2.3. Simulations with MOLA DTMs

MOLA is a laser altimeter [11] that measured Mars's topography for about a decade until 2006. It mapped the surface topography with a one meter radial resolution, and a 75 m spot size on the surface [12]. These data allowed for a nearly global Martian altimetry gridded product at a 463-m resolution around the equator and midlatitudes (Table 1) and 115 m around the poles. With our work being focused on the midlatitudes, we consider that all MOLA products are at the same midlatitude resolution. MOLA Digital terrain Models (DTMs) are found on the PDS and are the most used products for SHARAD data interpretation [3,13–15]. One of the reasons why MOLA data are widely used is the nature of the topographical measurement.

**Table 1.** Summary of the resolution of the different DTM sources used in this paper.

| Instrument | DTM Resolution (m/pixel) |
|---|---|
| MOLA | 463 |
| HRSC | 50–100 |
| CTX | 12–18 |
| HRSC DTM corrected by photoclinometry | 6 |

Laser altimeters measure the return time of a laser pulse and therefore provide us with an absolute measurement of the topography, knowing the orbitography of the spacecraft. This means that, while the low resolution may smooth some reflectors, MOLA DTMs do not create additional reflectors on the simulation. We will see that it is not necessarily the case for optical DTMs. Moreover, when studying regions that are relatively smooth, a higher resolution description of the surface increases the computing load of the simulation, without resolving more ambiguities. As shown in Figure 3, in the northern Martian plains, and in areas of low rugosity in general, when the reflectors are deep enough, the SHARAD simulations using MOLA products are precise enough that we can clearly identify clutter from subsurface echoes. We obtained the MOLA DTMs thanks to the PDS Geosciences Node's Mars Orbital Data Explorer.

When looking at shallow reflectors or areas of high roughness, however, a more detailed description is needed. In Figure 4, while the bulk of the echoes is clearly visible, the faintest details fail to appear in the simulation, therefore making the resolution of ambiguities impossible at this scale with MOLA DTMs.

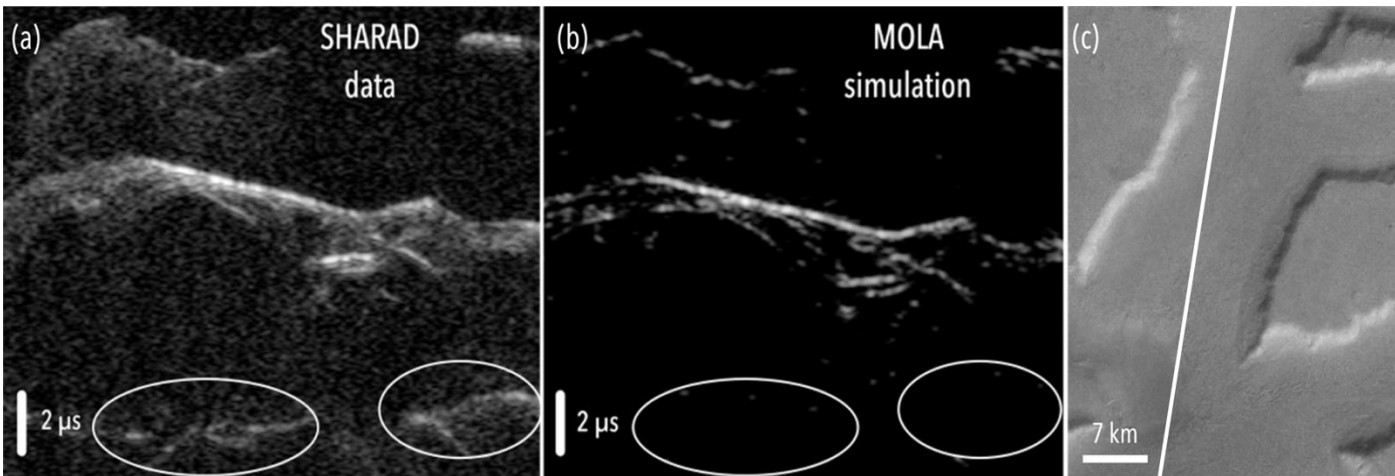

**Figure 3.** (**a**) A 40 km long portion of the SHARAD profile n°0806502 sounding Deuteronilus Mensae; (**b**) corresponding simulation using a MOLA DTM; (**c**) HRSC image of the terrain at nadir in the area where the radargram was taken. The two reflectors circled in white are present in the SHARAD data, but not in the simulation, meaning that they probably originate from the subsurface. Note that the two top echoes (left and right of the radargram) are off-nadir echoes originating from plateaus, thus arriving before the stronger nadir echo.

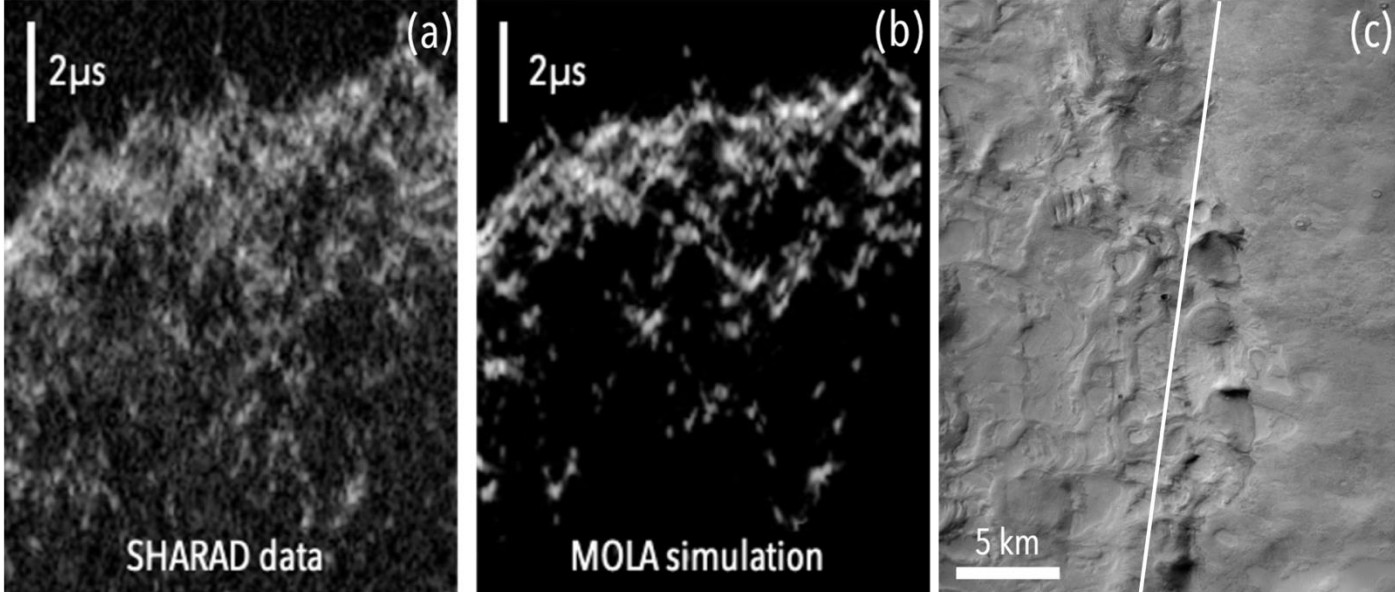

**Figure 4.** (**a**) A 40 km portion of the SHARAD radargram n°01708401 sounding Hellas Planitia; (**b**) corresponding simulation using a MOLA DTM. (**c**) CTX image of the terrain at the nadir on the area where the radargram was acquired. In areas where the roughness is higher, we can see that the identification of subsurface echoes is much harder with MOLA, as the faintest reflectors are not reproduced by the simulation.

### 2.4. Limits of MOLA DTMs for Shallow Subsurface Interpretations

In most of the southern midlatitudes, the terrain roughness is such that a MOLA description of the surface is not enough to yield satisfying results with simulations. Numerous subsurface reflectors could be misidentified and would therefore need further verifications to validate their presence. Surface clutter whose delay is similar to shallow subsurface reflectors comes from an area that is close to nadir. By geometrical construction, the result of the simulation is very sensitive to small errors in the topography around nadir, and a high-precision description of the surface is, therefore, needed to avoid misidentifications. Around Hellas Planitia, for example, dozens of subsurface reflectors have been

identified with MOLA simulations, and later discarded by higher resolution DTM simulations [16].

With our simulation tools, we used a MOLA DTM to simulate the surface echoes (Figure 5) on the SHARAD profile n°05128501, and indeed identified a reflector that is clearly visible on the SHARAD data and not in our synthesized radargram. Particularly when looking at shallow subsurface reflectors such as the one in Figure 5, a higher resolution description of the surface is needed, as surface and subsurface reflectors are more than ever mixed and complex to discriminate.

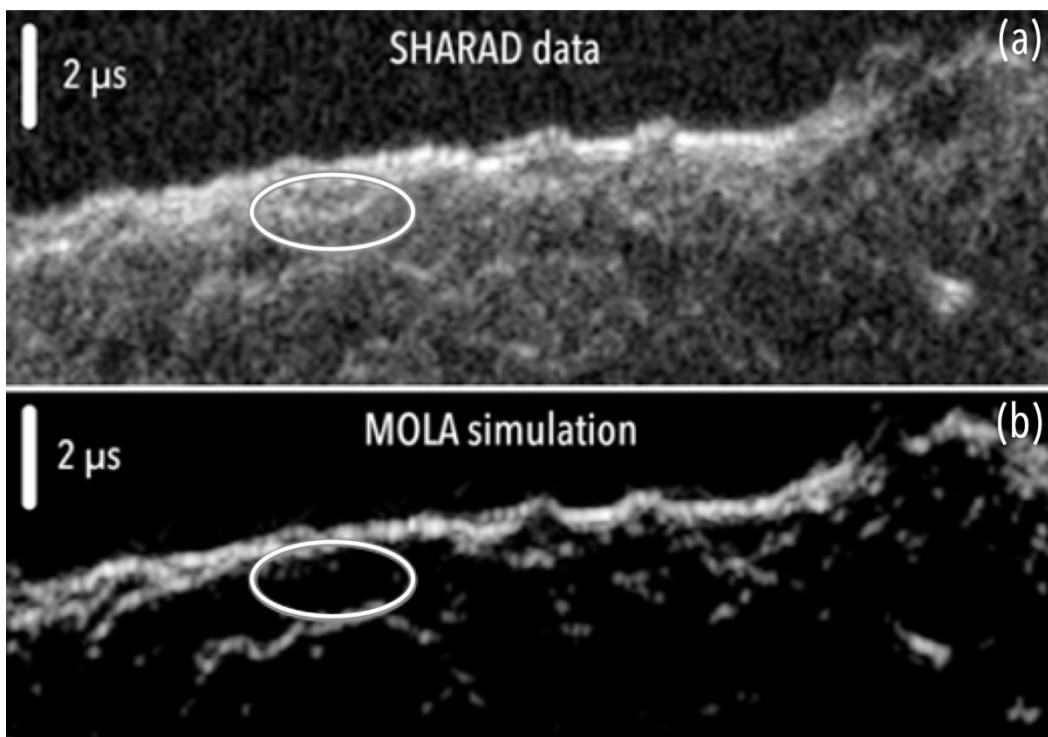

**Figure 5.** (**a**) A 65 km portion of the SHARAD profile n°5128501; (**b**) corresponding simulation using a MOLA DTM. The reflector circled in white is present in the SHARAD data but not in the MOLA simulation.

## 3. Extending the Radar Analysis Further with HRSC DTMs

As mentioned previously, we need a higher resolution DTM in order to properly simulate the signal, particularly in cases similar to Figure 5, where we want to look at shallow subsurface reflectors in rough areas. Optical DTMs on Mars are a great option because of their high resolution. All the optical terrain models on Mars have been obtained by photogrammetry, using a pair of images of the surface taken at different angles. These two images can be either taken at the same time or during two separate fly-bys of a given area, depending on the instrument (Table 2). One thing to note is that, on average, the higher the resolution of the instrument we choose, the smaller the DTM tiles will be and the less coverage percentage we will obtain. The optical DTM products with the highest coverage are products from the High-Resolution Stereo Camera (HRSC) instrument. We obtained these DTMs thanks to the PDS Geosciences Node's Mars Orbital Data Explorer. For our specific case, HRSC level 4 DTMs offer the best compromise between availability and resolution.

**Table 2.** Summary of the photogrammetric configurations for the instrument used for DTM generation in this paper.

| Instrument | Simultaneous Stereo Pair Acquisition | Convergence Angle (°) | Image Width at Closest Approach (km) |
| --- | --- | --- | --- |
| HRSC | Yes | 37.8 | 52 |
| CTX | No | 0–60 | 30 |

### 3.1. HRSC as an Upgrade of MOLA DTMs

HRSC uses its forward and aft-looking cameras for photogrammetry to generate DTM products at a resolution varying between 50 and 100 m per pixel (Table 1). As they provide a more detailed description of the topography, HRSC DTMs allow us to reproduce smaller surface echoes, therefore increasing the capability to discriminate between surface and subsurface reflectors. HRSC level 4 DTMs only cover about half of the Martian surface, which makes using them in the analysis impossible in some areas.

Where available, the use of HRSC to supplement the MOLA analysis has proven to be very efficient, as shown by [16]. They used SHARAD data to probe the region of Hellas Planitia for subsurface ice. They first used MOLA DTMs for their simulations and, upon comparison with the radar data, found many reflectors. To confirm their detection, they used higher resolution DTMs generated with HRSC images, resulting in about 100 m per pixel surface models. As HRSC DTMs do not cover all the identified reflectors, they could only confirm or discard part of the reflectors identified with MOLA. However, their conclusion to this confirmation is that more than 60% of the MOLA putative subsurface reflectors are invalidated by HRSC.

By visually comparing HRSC and MOLA DTMs (Figure 6), we can have a better grasp at the detail difference in a given area. As the resolution is better by a factor 4 to 8, finer details are visible, and the biggest ones become sharper. When using this HRSC DTM for a simulation with our tools, we obtained the result shown in Figure 7, and we can observe that finer details are indeed visible.

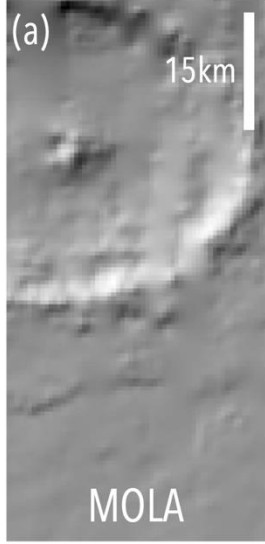 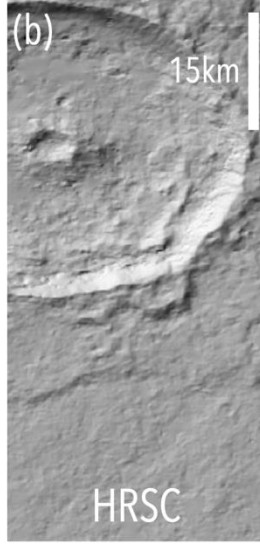

**Figure 6.** (**a**) MOLA DTM on Tarq crater at a 463 m per pixel resolution. (**b**) The corresponding HRSC DTM at 50 m per pixel.

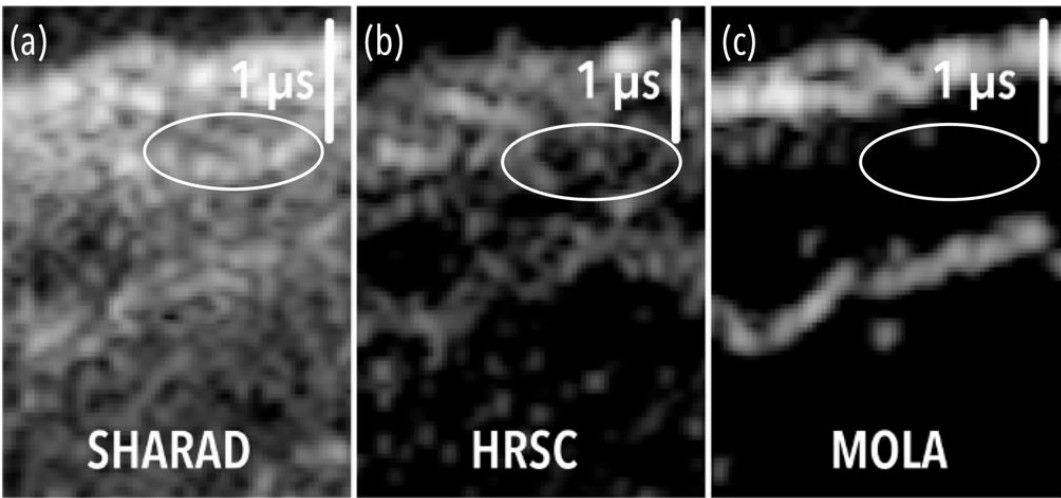

**Figure 7.** (**a**) A 12 km portion of the SHARAD dataset n°5128501 on Terra Cimmeria. (**b**) Simulation using a HRSC DTM. (**c**) Simulation using a MOLA DTM. Finer details are reproduced by HRSC, but a higher level of artifacts is present. The reflector circled in white is not visible in either of the simulations presented on the two rightmost images.

*3.2. Artifacts on DTMs Due to Photogrammetry*

Another aspect that catches the eye when comparing both simulations is that, in the HRSC simulation, the large reflectors are far less continuous than in the MOLA simulation. In addition, some bright areas of the HRSC simulation do not correspond to bright areas on the SHARAD data. Those two aspects will be referred to as "artifacts" in the following, because they are mainly due to small variations in the facets orientation at a small scale (a fraction of the Fresnel zone). These artifacts are due to the method of acquisition of the DTMs, which relies on matching pairs of pixels—each belonging to one image—and solving a system of collinearity equations involving the image coordinates of the matched pixels as observations, the 3D coordinates of the corresponding point on the scene, and camera parameters. The matching of pixels needs recognizable local intensity patterns on both pair images that are not guaranteed in the case of smooth terrains or when illumination conditions differ between the images. The precision of the reconstruction, therefore, highly depends on the contrast in the image and the orientation of the surface features relative to the sun. Consequently, photogrammetry introduces errors in the reconstruction of the surface, translating in small variability in the angles of the facets of the DTM and therefore producing these irregular patterns on the simulation. This makes the analysis much harder in some areas and is not ideal from the perspective of coming as close as possible to the SHARAD signal.

Using HRSC DTM n°h4209_0000_dt4, we performed a simulation with SPRATS (Figure 7), and did not see the reflector previously mentionned for the MOLA simulation (Figure 5). Given that an increase in resolution helped with the elimination of a large percentage of SHARAD reflectors identified with MOLA on Hellas Planitia that is similar in terms of roughness to Terra Cimmeria, performing simulations on the Tarq Carter area where the reflector was identified with a higher resolution DTM may help us to resolve this potential ambiguity. With their effective resolution of about 2 to 3 times and CTX pixel spatial resolution of 6 m (resulting in a 12 to 18 m resolution), CTX DTMs seem appropriate to describe the topography at a suitable scale for SHARAD simulations.

## 4. CTX as an Optimal DTM for SHARAD Data Interpretation

CTX is a camera onboard the Mars Reconnaissance Orbiter, producing 6 m per pixel images that are 30 km wide [17]. Using the information of two overlapping images taken during different orbits, one can reconstruct a DTM at a 12 m per pixel spatial resolution with photogrammetry (Table 1). The two images composing the stereo pair not being

acquired at the same time, locations where a DTM based on CTX can be made are limited to the mutual overlap of the CTX images. A given region might not contain the stereo pair needed for radar simulation, even though the CTX image coverage of the region is nearly 100%. The CTX DTMs that we used were generated using MarsSI [18].

*4.1. CTX Echo Reconstruction on a Rough Surface*

4.1.1. Resolution Higher Than the Radar Wavelength for a Quasi-Perfect Reconstruction of the Signal

When visually comparing an HRSC and a CTX DTM (Figure 8), we can see two main differences. First, finer details are visible on the CTX DTM, which is due to the increase in resolution, and second, the surface appears to be much smoother on CTX, meaning that we should have far fewer artifacts and more continuous reflectors on the simulations. The increase in resolution came with a decrease in the size of the artifacts, which is much better for the simulation, as will be discussed later.

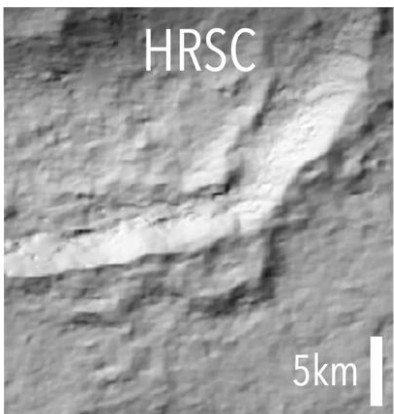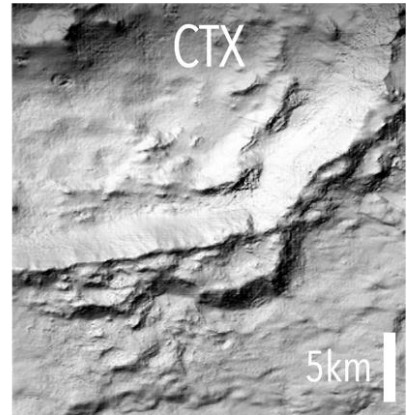

**Figure 8.** Visual comparison of the reconstruction of the edge of Tarq crater (southern midlatitudes) with a HRSC DTM at 50 m per pixel (**left**) and a CTX DTM at 12 m per pixel (**right**).

With the final resolution of CTX products being below SHARAD's wavelength, the level of description of the surface is the most appropriate for simulations, given the size of the DTMs and their resolution. While a higher resolution DTM should in theory be better for simulations, its use would result in a great increase in computing time. The span of the DTM would also need to be great enough to include some context.

When comparing the simulations with HRSC (50 m per pixel) and CTX (12 m per pixel) products with the SHARAD data (Figure 9), we can see that finer details are visible on the CTX simulation, as expected. Virtually no artifacts are visible, and even the faintest echoes seem to be well reproduced. This confirms the idea according to which a description of the surface at the scale of the wavelength or better is needed for fine radar data analysis with simulations. However, the ratio of the number of facets—and therefore of computing time—between MOLA and CTX is about 1500, which is not negligible. To provide an order of magnitude, a simulation taking 5 min for MOLA would take 5 days with CTX. To overcome this resolution issue, we can consider artificially reducing the resolution of the DTM.

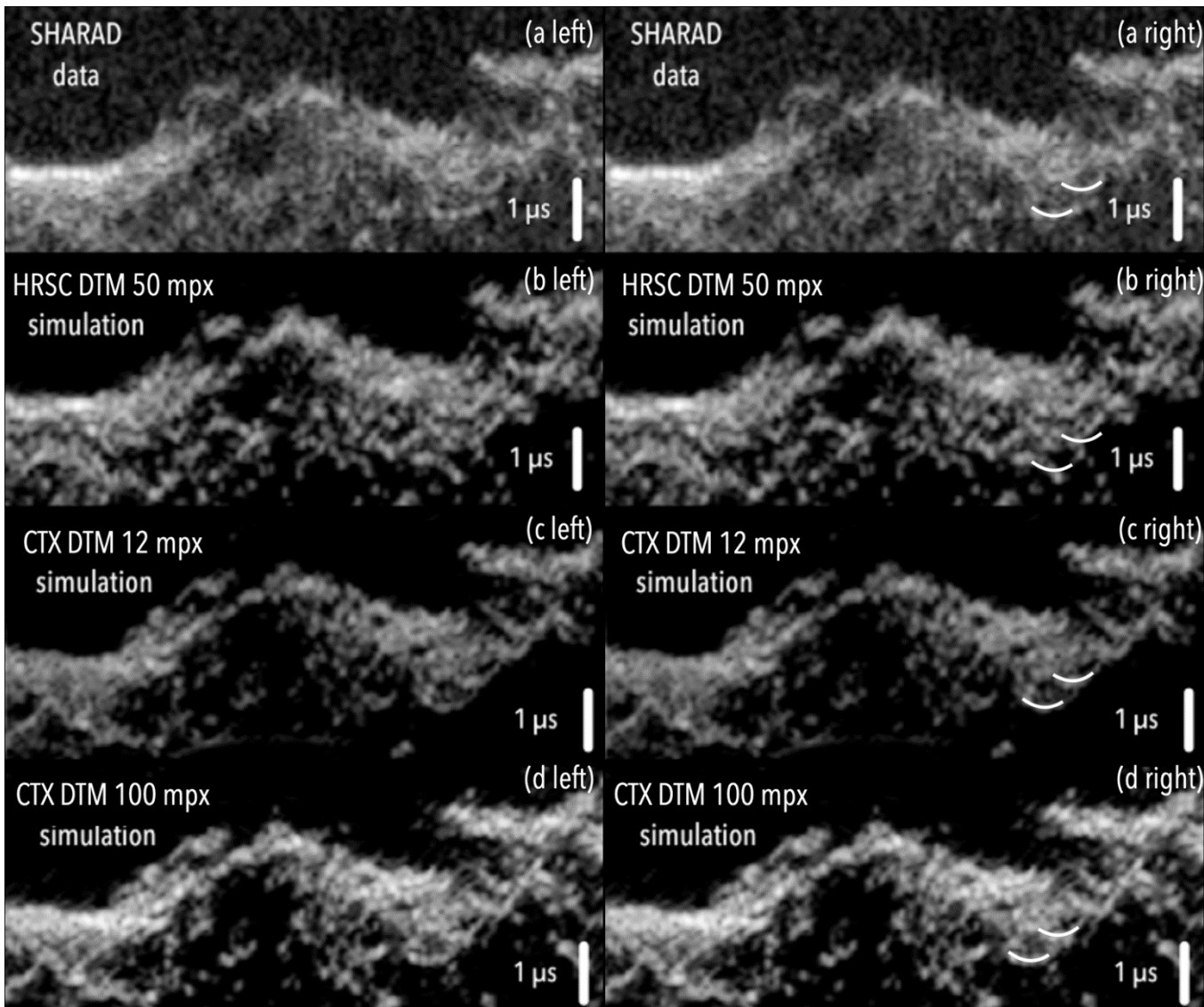

**Figure 9.** Comparison between a 40 km portion of the SHARAD dataset n°5128501 and the simulation of it using different DTMs. The results using the downsampled CTX DTM (**d**) at 100 m per pixel represents the echoes more accurately than the simulation using the HRSC DTM at 50 m per pixel (finer details visible and lower level of artifacts). Parabolas on the right images highlight the echoes that are most significatively improved by the CTX simulation, as the left images are there to visualize the echoes without figures on top.

### 4.1.2. Impact of the Resolution Reduction on CTX DTMs

When performing a simulation with a downsampled CTX DTM at 100 m per pixel, the simulation yields better results than the one made with a HRSC DTM at 50 m per pixel (Figure 9); small reflectors are better reproduced by the simulation using the CTX DTM. We can see it more clearly in the two reflectors highlighted by the parabolas on the right of Figure 9, which are not visible on the HRSC simulation compared to the CTX one. A straightforward explanation for this difference in results is that the order of magnitude of the artifacts found in the original CTX DTM are in the order of magnitude of the native resolution (12 m). Therefore, when downsampling the DTM, these artifacts are buried under the low resolution and smoothened. On the other hand, the artifacts found on the HRSC DTM are in the order of 50 m, much higher than those for CTX. When comparing the simulation made with the downsampled CTX DTM to the original CTX simulation, we can observe that, while the faintest details have either disappeared or merged together, most of the fine details visible on the simulation made with the 12 m per pixel DTM are

visible on the 100 m per pixel downsampled DTM. This result shows that, with a downsampled DTM at 100 m per pixel, the simulation yields similar or better results than even a native 50 m per pixel model, so it is therefore preferable to work with such DTMs when available.

*4.2. Results in the Terra Cimmeria Region of Interest*

Given the performance of the aforementioned downsampling method with a CTX DTM, it would be interesting to apply it to the reflector identified around Tarq crater to further investigate the origin of the non-detection on the HRSC simulation. However, only one of the two CTX images needed for the photogrammetry is available for the specific region we want to observe. As a first step, we mapped, on the available CTX image, the location of the identified echo supposing that it is coming from the surface (Figure 10). To achieve this, we measured the distance between the position of the spacecraft taken from SPICE and the position of the surface extracted from the HRSC DTM. We converted it into time delay and extracted the points of the DTM that correspond to the same delay as the echo identified by [10]. We then mapped a CTX image on top of the DTM to see more details on the terrain. We can clearly see that the calculated position of the echo matches the edge of a plateau. This might be another hint that the HRSC DTM was flawed at this specific place and that we should find a way to look further into it.

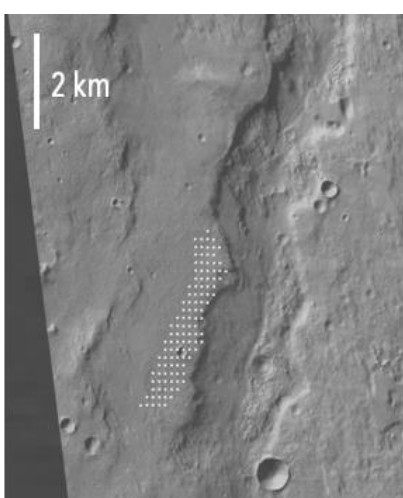

**Figure 10.** Mapping the identified echo on a HRSC DTM with a CTX image mapped onto it. The area of origin of the echo is materialized by the white dotted area in the center. If the echo came from the surface, it would come from the edge of a plateau.

4.2.1. Overcoming the Lack of CTX Coverage with Photoclinometry

Looking the topography of the region of interest from a HRSC DTM amplified by a factor of 20 (Figure 11), we can see that the edge of the plateau mentioned above is rounded by the photogrammetry while appearing quite abrupt on the CTX image (Figure 10), which still confirms our flawed DTM hypothesis due to photogrammetry limitations. Given that we only have a single image coming from CTX, but since we also have the information of the relative position of the sun and the spacecraft in relation to the scene at the time of capture of the image, we can apply a photoclinometry algorithm. It will convert the intensity variations of the CTX image into fine topographic quantitative details and merge the information with the coarse HRSC DTM [19]. The detailed flowchart of the processing for the generation of the DTM corrected by photoclinometry is presented Figure 12. The resulting fine DTM represented with the same exaggeration factor of 20 can be seen next to the original HRSC DTM (Figure 11), and it appears that the photoclinometry corrected the smoothed edge of the HRSC plateau. In the following, the corrected DTM

using photoclinometry is referred to as "HCPC" (HRSC-Corrected surface model using Photoclinometry with CTX).

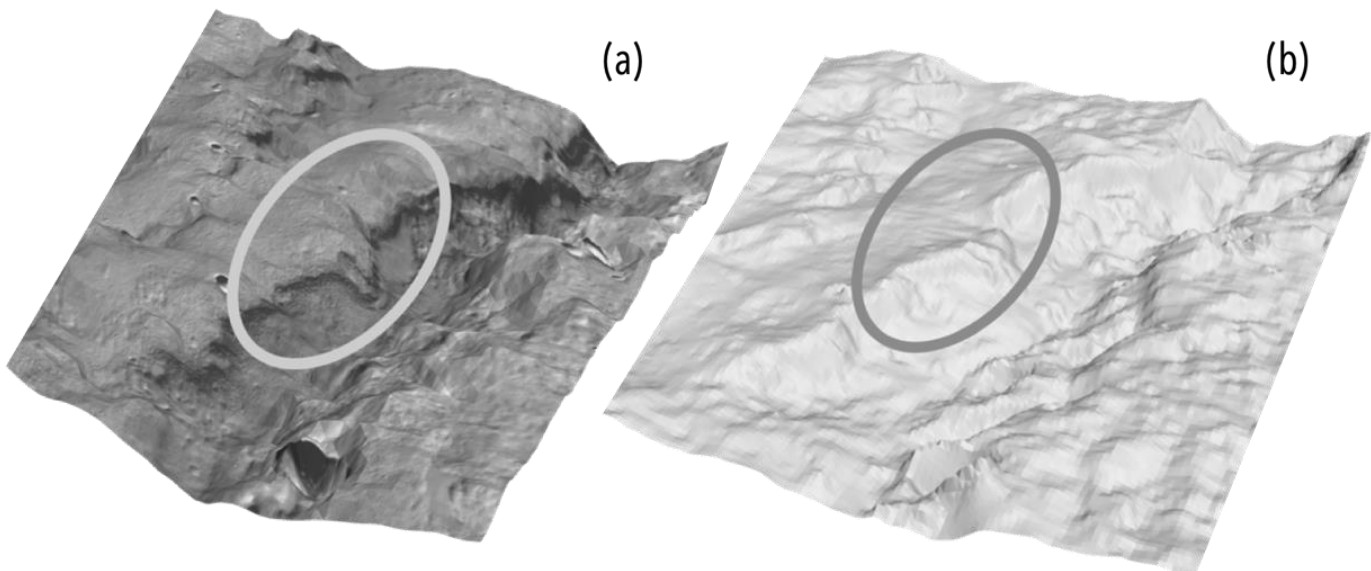

**Figure 11.** Comparison between vertical exaggeration of topographies (exaggeration factor of 20) for the HRSC DTM with a CTX image mapped on it (**a**) and the HCPC DTM (**b**). The edge of the circled plateau was straightened by photoclinometry.

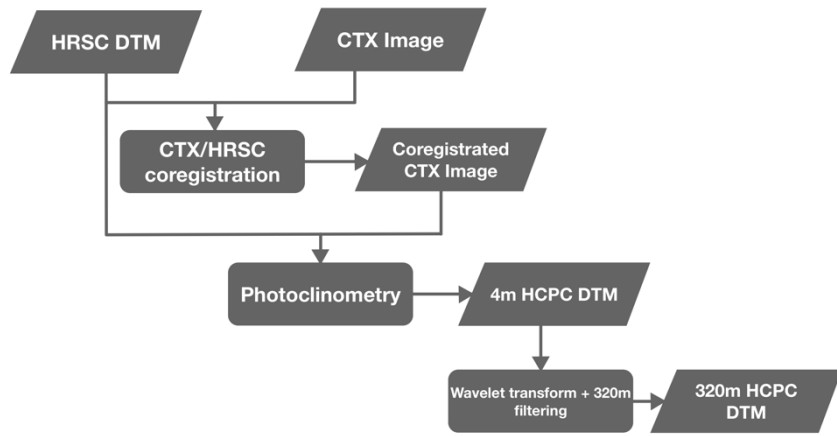

**Figure 12.** Data-processing flowchart for photoclinometry using the HRSC DTM and the CTX image.

In order to see the impact of the correction by photoclinometry, we observed the value of the angle of each facet of the DTM relative to the spacecraft, which provides us with a hint on where the radar signal could come from. Figure 13 represents the calculated angles of the facets from the uncorrected HRSC DTM compared to the same calculation made on the HCPC DTM. We notice that the plateau has indeed been straightened by the photoclinometry and that its edge appears to be leaning towards the spacecraft trajectory, which might return some radar signal during the simulation.

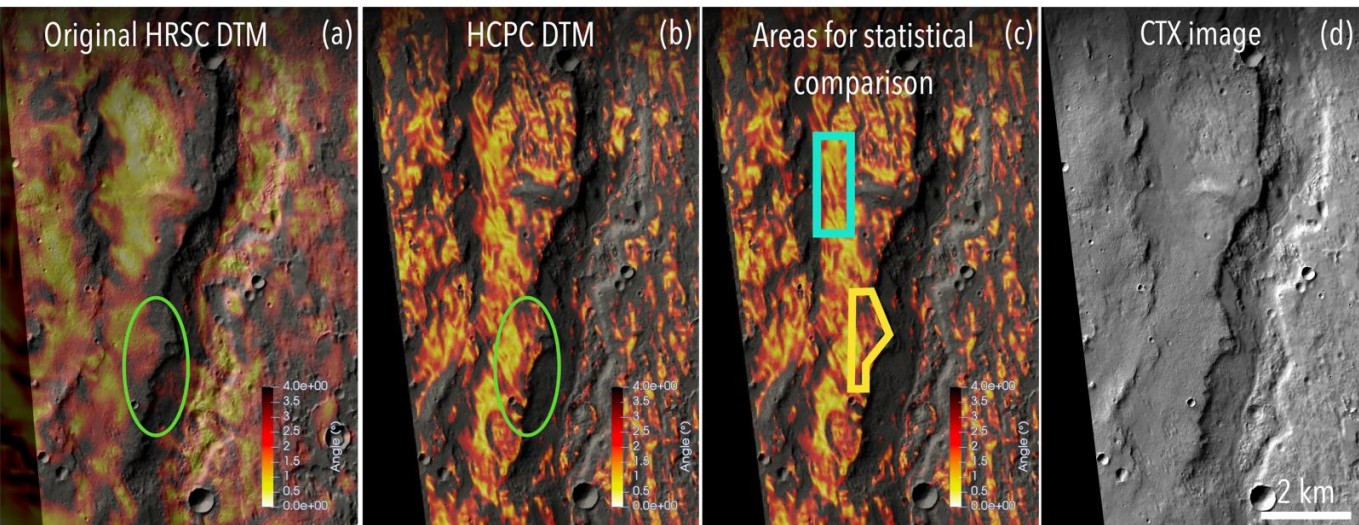

**Figure 13.** (**a**,**b**) Comparison of the angles between the facets and the spacecraft for the original HRSC DTM (**a**) and for the HCPC DTM (**b**). (**c**) Outlines of the two areas used for statistical comparison on the northern (cyan) and southern (yellow) plateaus. The southern plateau area geometry was chosen to match the calculations of the potential origin of the reflector (Figure 10). Both areas are about 1.5 km². (**d**) CTX image for context. The angles are shallower in the refined DTM in the area where the reflector is thought to come from (circled in green).

To see whether the correction of the plateau would be sufficient for it to result in a reflector on the simulation, we compared the statistics of the topography of the corrected plateau to a similar one north of it, which is indeed responsible of a reflection on the MOLA simulation (Figure 13c). We noticed that the statistics of both plateaus, namely the mean angle, are similar on the HCPC DTM (Table 3), but not on the MOLA DTM. This shows that both plateaus should have similar topographies relative to the spacecraft and that the southern plateau should also result in a reflector on the radargram. The issue with the HCPC DTM is that the variance in the angles is too great to result in a coherent reflector at a km scale. When only looking at the mean angle, however, an area of the size of the edge of the southern plateau (about 700 m by 1 km) should return a signal strong enough to be present on the simulated radargram, even with a 1° mean slope relative to the spacecraft.

**Table 3.** Comparison of the mean and variance of the incidence angles for MOLA and HCPC DTMs, along with the variance in range on the northern and southern plateaus (see Figure 14 for the detailed outlines of the area).

|  | **Northern Plateau** | **Southern Plateau** |
| --- | --- | --- |
| HCPC mean angle (°) | 1.05 | 1.31 |
| HCPC angle rms (°) | 3.24 | 4.36 |
| HCPC slant range rms (m) | 23.13 | 27.33 |
| MOLA mean angle (°) | 1.34 | 4.26 |
| MOLA angle rms (°) | 0.1 | 0.6 |
| MOLA slant range rms (m) | 20.30 | 26.61 |

4.2.2. High Sensitivity to Small-Scale Features for Radar Simulations

When performing a simulation with the HCPC DTM, we obtained the result shown Figure 14b. Some structures are recognizable, but most of them are surrounded by bright clutter, so the interpretation is not very clear. To overcome this issue, we performed a wavelet transform of the original DTM refined by photoclinometry and derived from it a DTM filtered at 320 m. The scale was chosen as a compromise between smoothening small-scale artifacts and still keeping a high enough spatial resolution to correctly

reproduce the plateau. This DTM has the advantage of filtering out the small-scale variations in the DTM, therefore greatly reducing the clutter present in the simulation (Figure 14c). By superimposing the simulation and the SHARAD data in a RGB composition (Figure 14d), we can see that the reflector identified in [10] is well reproduced by the simulation, confirming that the reflector is indeed coming from the plateau highlighted in Section 4.2.1.

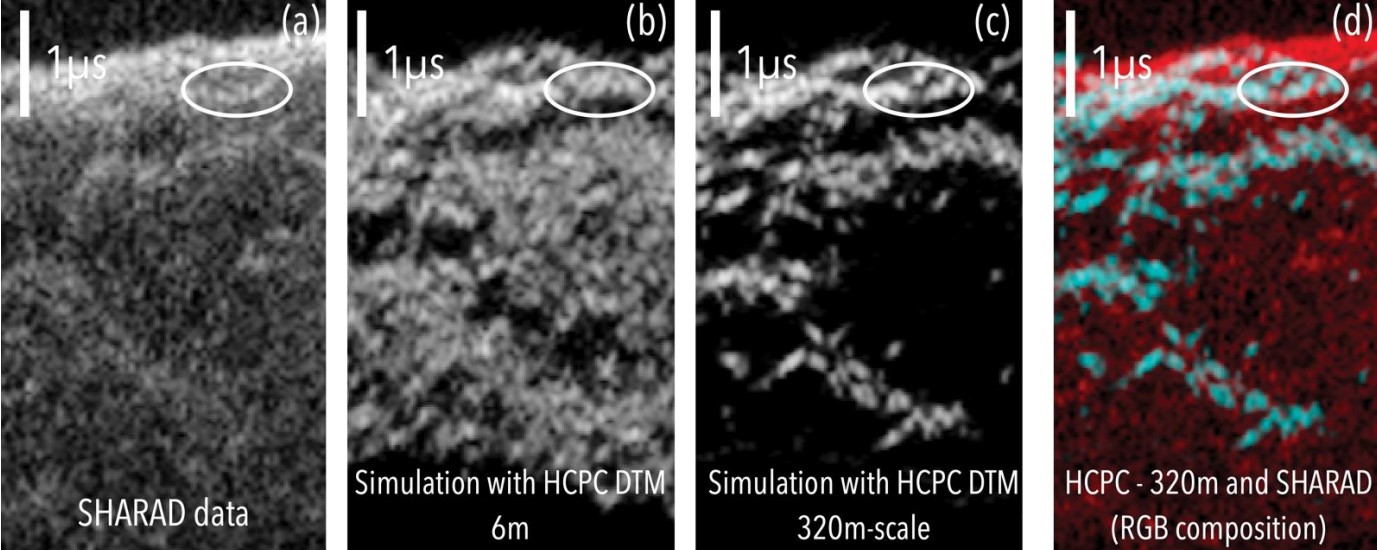

**Figure 14.** Comparison between a 18 km portion of the SHARAD dataset n°5128501 (**a**) and simulations using the corrected DTM with photoclinometry. The simulation in (**b**) was performed with the original DTM refined by photoclinometry at 6m. We do not see the surface on the simulated radargrams due to the DTM not covering the nadir part of the trajectory. (**c**) Wavelet-transformed DTM filtered at 320 m. (**d**) RGB composition with the SHARAD radargram on the red channel, and the simulation using the filtered DTM refined by photoclinometry on the cyan channel. The reflector that we are looking for is circled in white.

Figure 15 represents the evolution of the round-trip distance between the radar and the DTM across-track. The distance was plotted modulo lambda, so it can be seen as a phase evolution: areas of stationary phase are the areas where the echoes are coming from, while areas where the phase is quickly rotating are areas where we do not see any echo coming from. Using this visualization method allows us to translate why we see bright and continuous reflectors and little to no artifacts in MOLA simulations, and why we see a lot of artifacts in the 6 m DTM. We can clearly see in Figure 15c that the filtered photoclinometry DTM at 320 m is a compromise between MOLA and the 6m DTM; while the small-scale asperities were smoothened (see the center of the image), the plateau and other smaller-scale features were corrected. This allows for the echo coming from the plateau to appear clearly in the simulation using the 320 m DTM. The result of the simulation using the 6 m DTM (Figure 14b) shows that radar simulations at SHARAD's wavelength are highly sensitive to small-scale variations on DTMs and that one should be cautious when interpreting radar simulation made with high-resolution DTMs.

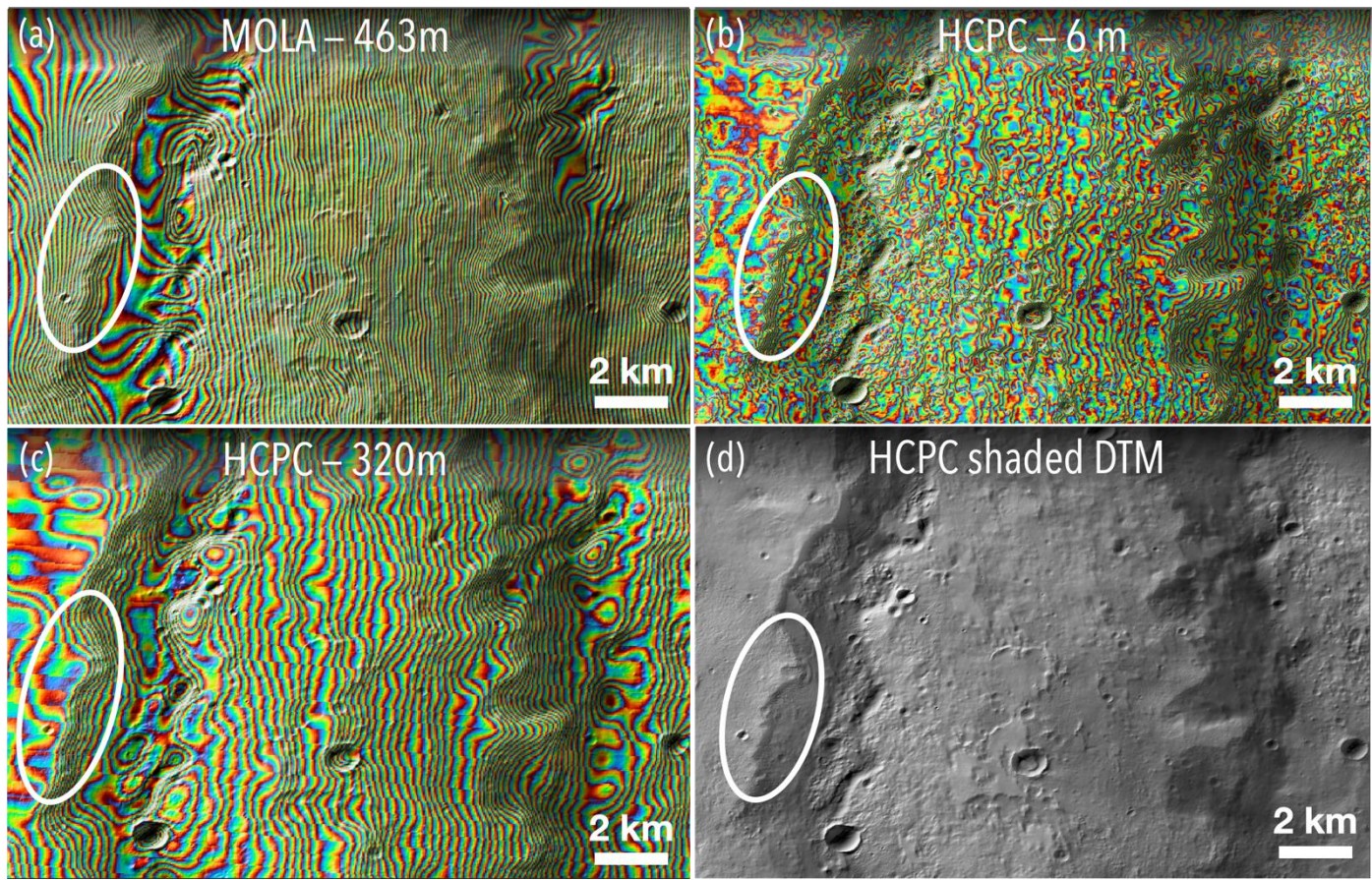

**Figure 15.** (**a–c**) Mapping of the across-track round-trip distance between the points of the DTMs and the spacecraft. The slant range was mapped modulo lambda using a cyclic color scale that can be directly linked to the evolution of the signal's phase across track. The plateau that we want to correct by photoclinometry is located in the white circle. The areas where the phase is stationary are the areas where the radar signal reflects on, and the larger the area, the brighter the echo. (**a**) Range map in the MOLA DTM. (**b**) Range map in the 6m HCPC DTM. (**c**) Range map of the 320 m scale wavelet-transform of the HCPC DTM. (**d**) HCPC-shaded DTM to visualize the terrain details.

### 4.3. Detection of DTM Errors with Radar Simulation

In particular, the above study showed that by performing a radar simulation and comparing it with the original data, we are able to detect small-scale artifacts. In fact, DTM-isolated artifacts, at a scale comparable to a fraction of the Fresnel zone, create disturbances in the facet angles and, thus, discontinuities and intensity biases in the radargram, while the overestimation of the topographic variance at even smaller scales (6–50 m) by photoclinometry destroys the phase coherency of the radar signal in some places, masking real echoes.

It also showed that optical DTMs, while convenient for their high resolution and relatively high coverage compared to laser altimetry, can include small- and large-scale artifacts, which are responsible for errors in the radar data interpretation. Radar simulation, therefore, proves to be very efficient to assess DTM quality, which is necessary for other missions, especially those where no laser altimeter is present onboard to precisely measure the topography and to use as a reference.

### 5. Conclusions

In this paper, we showed that while SHARAD simulations using low-resolution DTMs such as MOLA are practical to detect deep and bright reflectors, for shallow reflectors (and therefore those detected close to nadir), the simulation has a great sensitivity to small errors in the inclination of the facets. A precise reproduction of the surface is

therefore needed. For that purpose, we showed that high-resolution DTMs acquired by photogrammetry are effective notwithstanding certain limitations: they are usually scarcer in surface coverage and induce several fold increases in computing time for radar simulations. Another drawback to these high-resolution DTMs is related to their acquisition method: photogrammetry is only an estimation of the topography and can generate artifacts that are highly dependent on the scene, and therefore are hard to predict and characterize, as we saw with the plateau near the Tarq crater and the HRSC DTM. These issues are not present on laser altimetry DTMs. We also showed that simulations are very sensitive to small-scale errors on the surface down to scales of a fraction of the Fresnel zone, which may make the analysis difficult, except if filtering (e.g., a wavelet transform, for our case) is performed on the DTM before radar simulation. Radar simulations at SHARAD's wavelength must be performed while keeping in mind that DTMs can be biased, and using multiple analysis techniques when looking at small and shallow reflectors is a way to reduce the uncertainty on their identification.

We also highlighted the fact that photoclinometry [19] is a technique that allows us to partially correct those biases and to eliminate wrongfully identified subsurface echoes. Thanks to this technique, we confirmed that the reflector identified by [10] is, in fact, due to a surface feature.

This study also showed that, conversely, the comparison between radar and simulated data provides a way to assess the quality of DTMs, which is crucial in missions where there is no laser altimeter with which to compare optical DTMs, such as the Europa Clipper/REASON [20] and JUICE/RIME radars [21].

**Author Contributions:** conceptualization, L.D., A.H., W.K. and S.Z.; methodology, L.D., S.D., A.H., W.K. and S.Z.; providing of photoclinometry-corrected DTMs, S.D.; advising on optical DTMs, S.D.; investigation, L.D.; writing—original draft preparation, L.D.; writing—review and editing, S.D., A.H., W.K. and S.Z.; supervision, A.H. All authors have read and agreed to the published version of the manuscript.

**Funding:** This research received no external funding.

**Data Availability Statement:** MOLA and HRSC DTMs can be acquired via the PDS and the CTX DTM via MarsSI [18]. The DTM corrected by photoclinometry can be acquired upon request. The simulated datasets used in this paper can be acquired upon request.

**Acknowledgments:** This project is supported by CNES. The SHARAD instrument was provided to the NASA's Mars Reconnaissance Orbiter (MRO) mission by the Italian Space Agency (ASI), and its operations are led by the DIET Department, University of Rome "La Sapienza" under an ASI contract. CTX DTM data were processed with the MarsSI (marssi.univ-lyon1.fr) application founded by the European Union's Seventh Framework Program (FP7/2007-2013) (ERC Grant Agreement No. 280168).

**Conflicts of Interest:** The authors declare no conflict of interest.

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
