# Peer review of "Resolving Ambiguities in SHARAD Data Analysis Using High-Resolution Digital Terrain Models"

_remotesensing, doi:10.3390/rs15030764_

Round 1

Reviewer 1 Report (Previous Reviewer 1)

Author Response

Dear reviewers, 

Thank you for your comments on the manuscript, I believe they have greatly improved its quality and clarity. We modified our manuscript according to your comments and we hope it suits everyone. As was suggested by one of the reviewers, we had the english checked via the MDPI service.

Thank you, 

Léopold Desage on behalf of the authors

Reviewer 2 Report (Previous Reviewer 3)

I don't have further comments

Author Response

Dear reviewers, 

Thank you for your comments on the manuscript, I believe they have greatly improved its quality and clarity. We modified our manuscript according to your comments and we hope it suits everyone. As was suggested by one of the reviewers, we had the english checked via the MDPI service.

Thank you, 

Léopold Desage on behalf of the authors

Reviewer 3 Report (Previous Reviewer 2)

Dear Authors,

Good effort to include Clinometry for HRSC products / DTM refinement. The following suggestions can be considered for the improvements:

1. Flowchart(s) can be more detailed and prepared correctly. In Figure 11, "photoclinometry" should be a lateral input rather than in the main pipe line producing the fused/corrected output in the main pipe line/process flow. Flowchart can also be suitably placed as per the standard Methodology section to have more insight/description in the correct place.

2. The manuscript needs to be prepared as a technical document. The conclusion needs to be rewritten. Thanks, the part should only be in the acknowledgment. 

best wishes,

Author Response

Dear reviewers, 

Thank you for your comments on the manuscript, I believe they have greatly improved its quality and clarity. We modified our manuscript according to your comments and we hope it suits everyone. As was suggested by one of the reviewers, we had the english checked via the MDPI service.

Thank you, 

Léopold Desage on behalf of the authors

Round 2

Reviewer 3 Report (Previous Reviewer 2)

Good.

best wishes,

Author Response

Thank you, 

Léopold Desage on behalf of the authors

This manuscript is a resubmission of an earlier submission. The following is a list of the peer review reports and author responses from that submission.

Round 1

Reviewer 2 Report

Dear Authors,

Topic is of great scientific interest. Manuscript need major grammatical and technical writing corrections. Proper sections on Data/Material and Methodology needed with flowchart on Methodology. Sections on Results and Discussions shall also be used for the right style of Journal manuscript. Correct the terminologies throughout the manuscript especially for spatial resolutions/artefacts and reasons for discripencies highlighted in the study.

1.       Abstract: reframe “The SHAllow RADar (SHARAD) onboard Mars Reconnaissance Orbiter (MRO) ….fathoms…. the first hundreds of meters of the Martian subsurface at a frequency of 20 MHz.”

2.       Line(s) 52/58/61/62/92/100/107/115……/148/150/159/163/164/170/171/…190/193/236/245/ 264/…/ 291/300/301/303/315/317/324/328/346/358/359/364/365/366 / 403/406/409/426/429/431: rewrite sentence(s) appropriately/technically without using we/I/you/our/In/by/For(avoid punctuations for beginning a new sentence)….

3.       Make captions of figures more effective for communication of information/interpretation.

4.       Line 61: replace “Our aim is to simulate…” by “The study aims at….”.

5.       Line 87: reframe sentence “For the case of SHARAD, the SAR synthesis 87 of a simulation using a MOLA model at 463m is shown Figure 1.b.”

6.       Lines 175-177: Make the correction procedure clear – “…o eliminate wrongful interpretations”

7.       Line 185:  reframe sentence “…Following that, …..”

8.       Line 208: instead of writing: - ”… photogrammetry 208 introduces errors…” mention the factors like contrast in Stereopairs and other relevant factors responsible for such issues in photogrammetry in the procedures/methodology used in study.

9.       Line 213: reframe sentence: “…In their 2019 paper….” Suitably.

10.   Line 222/234/236: correct: - pixel scale (6 meters) / 12m (line 236) or pixel spatial resolution. Choose correct terminologies.

11.   Lin 229” correct “…rightern images…”

12.   Line 253-255: reframe: “While a finer model should in theory be better for simulations, 253 the computing time would skyrocket and the span of the model would need to be great 254 enough to include some context”

13.   Line 261-262: Reframe suitably- “…therefore in 261 computing time — between MOLA and CTX is about 1500, therefore not negligible.”

14.   Section 4.2.1. Overcoming the lack of CTX coverage with photoclinometry can have a clear flowchart.

15.   Figure 10: do you mean vertical exaggeration by “amplified topographies”. Use clear terminologies.

16.   Line 371-372: Write clearly about 320m model/ 6m model

17.   Line 439: may be shifted / mentioned in literature review/introduction section/ part “…Europa Clipper/REASON and 439 JUICE/RIME radars…”

18.   Conclusion need rewriting focusing on clear concluded points in properly formed paragraphs. More relevant references can be added...

Thanks.

Best wishes,

Reviewer 3 Report

see comments attached
